# The Molecular Mechanisms of Liver Fibrosis and Its Potential Therapy in Application

**DOI:** 10.3390/ijms232012572

**Published:** 2022-10-20

**Authors:** Danyan Zhang, Yaguang Zhang, Bing Sun

**Affiliations:** 1School of Life Science and Technology, Shanghai Tech University, Shanghai 201210, China; 2State Key Laboratory of Cell Biology, Shanghai Institute of Biochemistry and Cell Biology, Center for Excellence in Molecular Cell Science, Chinese Academy of Sciences, University of Chinese Academy of Sciences, Shanghai 200031, China

**Keywords:** liver fibrosis, cellular signaling pathways, antifibrotic compounds, natural compounds, drug delivery system

## Abstract

Liver fibrosis results from repeated and persistent liver damage. It can start with hepatocyte injury and advance to inflammation, which recruits and activates additional liver immune cells, leading to the activation of the hepatic stellate cells (HSCs). It is the primary source of myofibroblasts (MFs), which result in collagen synthesis and extracellular matrix protein accumulation. Although there is no FDA and EMA-approved anti-fibrotic drug, antiviral therapy has made remarkable progress in preventing or even reversing the progression of liver fibrosis, but such a strategy remains elusive for patients with viral, alcoholic or nonalcoholic steatosis, genetic or autoimmune liver disease. Due to the complexity of the etiology, combination treatments affecting two or more targets are likely to be required. Here, we review the pathogenic mechanisms of liver fibrosis and signaling pathways involved, as well as various molecular targets for liver fibrosis treatment. The development of efficient drug delivery systems that target different cells in liver fibrosis therapy is also summarized. We highlight promising anti-fibrotic events in clinical trial and preclinical testing, which include small molecules and natural compounds. Last, we discuss the challenges and opportunities in developing anti-fibrotic therapies.

## 1. Introduction

Long-term viral infections (HBV, HCV), alcohol abuse, fatty diets, autoimmune disorders, etc., are responsible for chronic liver disease that progresses from hepatic steatosis to inflammation, fibrosis, cirrhosis, and eventually hepatocellular carcinoma. Liver fibrosis has a significant influence on the morbidity and mortality of people with liver disease. Cirrhosis is associated with a higher risk of death and a higher incidence of hepatocellular cancer. Based on previous research, it is feasible to relieve or reverse the progression of liver fibrosis with, for instance, drugs to treat HBV and HCV that can reverse fibrosis in patients. However, there are currently no FDA and EMA-approved therapies to directly treat liver fibrosis. Understanding the pathogenesis and mechanisms of liver fibrosis and its clinical implications is vital to developing new antifibrotic therapies. Various cells and cellular signaling pathways are engaged in the mechanisms of liver fibrosis. Chronic liver disease can start with hepatocyte injury and inflammation associated with the production of cytokines and chemokines, such as TGF-b, IL-6, and TNF-a, then activate HSCs. The HSCs serve as the major cell type in the progression of liver fibrosis; its activation is intimately related to collagen synthesis and ECM accumulation.

The anti-fibrosis agents include lifestyle modification, bariatric surgery, and pharmacologic treatments, which include small molecular inhibitors, proteins, antibodies, and natural compounds. Targeting liver lipid metabolisms and oxidative stress, targeting liver inflammation and cell death, and targeting liver fibrosis are all examples of clinical trials and pre-clinical testing that focus on different characteristics of liver fibrosis. A THR-b agonist called Resmetirom was evaluated in a Phase III trial, and the latest results showed that it was effective in patients with NAFLD. Patients with F2–F3 fibrosis will be enrolled in the advanced research.

In conclusion, the complexes of the pathogenesis and mechanisms suggested combination therapies that target two or more strategies may be needed. Along with chemical compounds, more and more natural products have interestingly been prescribed for the treatment of liver fibrosis. The mechanisms of action for most natural substances are complex and varied, but they have shown promise in pharmacological activity in the treatment of liver fibrosis. Anti-fibrotic drug research, on the other hand, is fraught with challenges. Cell-specific targeting is an alternative strategy for increasing medication efficacy via receptor engagement, but it requires low immunogenicity and an efficient drug delivery system. In the following, we will briefly address the pathogenic mechanisms of liver fibrosis and signaling pathways involved, as well as various molecular targets for liver fibrosis treatment, especially promising anti-fibrotic events in clinical trials and preclinical testing, which include small molecules and natural compounds. Last, we will discuss the obstacles and opportunities that come with developing anti-fibrotic therapies.

## 2. Molecules and Mechanisms of Liver Fibrosis

Repeated and persistent liver injury, such as HBV and HCV infections, alcohol addiction, gut microbiota, parasites, autoimmune disorders, etc., causes liver fibrosis. Antiviral medications have made significant progress in avoiding the development of liver fibrosis in recent years, but the patients with hereditary or autoimmune liver disease, particularly nonalcoholic steatosis, remain to be cured with alternative treatments. Understanding the molecules and mechanisms in the development of liver fibrosis, including cell–cell crosstalk and signaling pathways could identify the potential targets in clinical treatment.

### 2.1. Cell Types in Liver Fibrosis

#### 2.1.1. Hepatocytes

There are two main cell types in the liver, parenchymal cells and non-parenchymal cells. As the dominant component of the liver, parenchymal hepatocytes occupied 60% of the total cell numbers and about 80% of the total liver volume. Hepatocytes play a crucial role in energy metabolism, detoxification, and protein synthesis. The development of NAFLD to NASH and liver fibrosis is closely associated with a series of liver injuries brought on by lipotoxicity, oxidative stress, ER stress, and inflammation that occur sequentially in different liver cells (hepatocytes and Kuffer cells) and which, in turn, activate liver regeneration and fibrogenesis. Regarding NASH-related liver fibrosis, the homeostasis of lipid and glucose metabolisms is critical for hepatocytes. As illustrated in Figure 1, the pathophysiology of NASH is dependent on the synthesis and disposal of free fatty acids (FFA). Fats from dietary are stored in adipose tissue (AT). Then, the lipolysis of triglyceride is transported to the liver via the blood [1]. Additionally increased visceral AT (VAT) is associated with increased insulin resistance and inflammation. De novo lipogenesis (DNL), which is mostly fueled by dietary carbohydrates in the intestine, is another important source of free fatty acids in the liver. In hepatocytes, FFA undergoes mitochondrial b-oxidation and re-esterification to generate TG, which is either exported into the bloodstream as VLDL or converted into lipid droplets and returned to the hepatocyte FFA pool [2]. When the disposal of FFA is overwhelmed, the over-accumulation of FFA can cause lipotoxicity, while also inducing ER stress, oxidative stress, and inflammation. All of these mechanisms lead to hepatocyte damage by releasing cytokines and chemokines such as IL-1b, IL-6, TNF-a, TGF-b, IL-18, and so on. Therefore, hepatocyte targeting is an alternative strategy in clinical application, which efficiently improves drug accumulation and lowers immunological responses. Most of the hepatocyte targeting is based on the specific receptors expressed on it, such as the asialoglycoprotein receptor (ASGPR), LDLR, and the folate receptor. Sorafenib loaded with folate receptor-targeted polymer lipid hybrid nanoparticles was formulated to enhance the therapeutic efficacy [3]. Sorafenib coated with folate-conjugated lipid (with chitosan surface nanoparticle) possesses great interaction with cancer cells compared with the non-targeted control group.

#### 2.1.2. Inflammatory Cells

Inflammatory cells are drawn in by released fibrogenic cytokines and chemokines, which can activate HSCs and MFs. Neutrophils are frequently dispatched to the injured liver as the first line of defense against apoptotic hepatocytes. In response to the injury, NKT cells and proliferating bile ductular epithelial cells (BEC) also activate and secrete inflammatory cytokines, which induce hepatic fibrogenesis. Furthermore, the production of TGF-b can in turn activate macrophages and HSCs, and bone marrow-derived macrophages, rather than liver-resident macrophages (Kuffer cells), are thought to be the main source of TGF-b. Macrophages play a dual role in the progression and resolution of liver fibrosis [4]. In the development of fibrosis, macrophages recruit more inflammatory cells to the damaged liver, stimulate inflammation by producing more cytokines and chemokines, and subsequently activate HSCs in the formation of fibrosis. Furthermore, during the remission of liver fibrosis, macrophages increase the secretion of metalloproteinases (MMPs), such as MMP9 and MMP2, while decreasing the production of fibrogenic and inflammatory substances [5]. The specialized macrophages, known as Kuffer cells ($CD11b^{+}$, $F4/80^{+}$, $CD68^{+}$, $CXCR1^{-}$), are essential for the liver’s systemic response to pathogens. Kuffer cells (KCs) can be activated by various damage-associated molecular patterns (DAMPs, such as free DNA, ATP, high-mobility group box 1) and pathogen-associated molecular patterns (PAMPs), Toll-like receptors (TLRs), and the P2X7 receptor. Interleukin-1b (IL-1b), IL-18, and other pro-inflammatory cytokines and chemokines (in particular CC-chemokine ligand 2 or CCL2) are released as a result of the activation in KCs and inflammasome construction. CCL2 and other cytokines encourage the recruitment of $CCR2^{+}$/ $Ly6C^{hi}$ monocyte into the wounded liver in mouse models, where they mature into inflammatory and fibrogenic LY-$6C^{+}$ macrophages. TGF-b1 and PDGF are the most important mediators released by these cells, which activate HSCs or portal/perivascular fibroblast into collagen-producing MFs. Further, they can promote nuclear factor kappa B (NF-kB), which in turn attracts MFs via chemokines, such as CCL2, to boost the survival of MFs [6]. The promotion of macrophage from pro-regenerative M2 to inflammatory M1 leads to hepatic injury induction and liver restoration improvement. The expression of SYK is induced in the M1 macrophage; the specific inhibitor of SYK pathway can significantly reduce inflammation. A PLGA nanoparticles-based delivery of SYK pathway inhibitor (R406), exhibits stronger ameliorated steatosis, inflammation, and fibrosis [7].

#### 2.1.3. Hepatic Stellate Cell

In the normal liver, HSCs are found in the subendothelial area of the Disse and have close connections with the surrounding hepatocytes and sinusoidal endothelial cells (SECs). In addition to being able to function as pericytes specific to the liver, HSCs are physiologically responsible for the synthesis and remodeling of ECM in the Disse area and for the storage and metabolism of vitamin A and retinoids. Following stimulation with fibrogenic mediators, HSCs become activated, transdifferentiating from vitamin-A storing cells to myofibroblasts with upregulation of a-smooth muscle actin and enhanced ECM production [8]. Activated HSC can be induced by a variety of molecules and signaling pathways. On the one hand, HSC receptors and related signaling pathways are in charge of detecting external signals that control the activation or quiescence of HSCs through interactions with certain ligands. Importantly, key fibrogenic and proliferative pathways, such as transforming growth factor-b (TGF-b), platelet-derived growth factor (PDGF), vascular endothelial growth factor (VEGF), and connective tissue growth factor (CTGF), all function via particular receptors to contribute to advanced hepatic fibrogenesis. Platelets activate HSCs and promote liver fibrosis mainly by producing TGF-b and PDGF-b. The most potent profibrogenic cytokine, TGFb, activates HSCs either SMAD2/3-dependently or independently. The TGFb targets in mouse-activated HSCs were found to be the Col1a1, Col1a2, Activin, and Pai1 genes. Through JAK-STAT3 signaling, leptin, IL-6, and IL-17 can also activate Col1a1 transcription. Additionally, Col1a1 expression in activated HSCs is promoted by CTGF and IL-13 via a TGF-b1 independent mechanism. HSC activation has been linked to the activation of immune-related signals and cytokines by toll-like receptors (TLRs, TLR4, TLR9, TLR3), DAMPs, and LPS. Furthermore, adipokines facilitate tissue communication among the adipose, liver, and other tissues. HSC activation is aided by the Hedgehog (Hh) ligand and its receptor smoothened homolog (SMO). G protein-coupled receptors (GPCRs) expressed by HSCs, including cannabinoid receptor 1 (CB1), CB2, proteinase activated receptor 2 (PAR2), 5 hydroxytryptamine receptors (5 HTs), GPR1 (also known as succinate receptor 1), C–C chemokine receptors (CCRs), and type 1 angiotensin II receptor (AT1R), interact with chemokines and angiotensin to either adversely or positively alter HSC activation. Diverse nuclear transcription factor receptors that are expressed on HSCs, including liver X receptor (LXR), farnesoid X receptor (FXR, also known as bile acid receptor), PPARg, PPARd, vitamin D receptor (VDR), and nuclear receptor subfamily four group A member one (NR4A1) can negatively modulate HSC activation and fibrosis progression by regulating glucose and lipid metabolism [9].

Changes in epigenetics of HSCs including DNA methylation, miRNAs, and histone modification, on the other hand, also control both activation and inactivation of HSCs, making them attractive targets for hepatic fibrosis treatment. Gene expression is reduced when cytosine-phosphoguanine (CpG) methylation is in promoter regions. Methyl-CpG binding protein 2 (MECP2) and histone-lysine N-methyltransferase enhancer of zeste homolog 2 (EZH2) can promote HSC activation and fibrosis by repressing PPARg transcription. Small non-coding RNAs called miRNAs (22 nucleotides in length) control mRNA degradation to regulate post-transcriptional gene expression. In the progression of liver fibrosis, profibrogenic miRNAs include miR-221, miR-222, miR-27, and others, while antifibrotic miRNAs include miR-214, miR-29, miR-378a, miR-148a, and others [10]. To selectively inhibit the activated HSCs, miR-29b was combined with a natural product, germacrone (GMO) co-encapsulated into nanoparticles (NPs, based on PEG-PLGA) and GMO, and miR-29b loaded with NPs exhibited great cytotoxicity to activated HSCs [11]. Additionally, based on those receptors that are specifically over-expressed on activated HSCs, receptor-mediated drug targeting has been reported to improve the therapeutic effect of antifibrotic drugs. For example, AT1 receptor blocker, losartan, was loaded with hyaluronic acid (HA) micelles to attenuated HSC activation. A stronger increase delivery rate of losartan was observed in the losartan-HA group versus losartan alone [12].

#### 2.1.4. Myofibroblasts

Myofibroblasts (MFs) are absent in a healthy liver. When the liver is injured, myofibroblasts are activated, which results in high collagen synthesis and extracellular matrix protein (ECM) accumulation, mainly via downregulation of matrix metalloproteinases (MMPs, such as MMP-1, -3, -8, -9, and -13) and upregulation of the tissue inhibitors of MMPs (TIMPs, such as TIMP1 and TIMP2). It is difficult to identify the origin of MFs in clinical disease, while the experimental models of liver fibrosis indicate that more than 90% of the cells that produce collagen are activated HSCs and portal fibroblasts. Several other sources of hepatic MFs have been suggested, including epithelial-to-mesenchymal transition (EMT) and bone marrow-derived mesenchymal progenitors (or MSCs) [13]. The shift of mature epithelial cells into fully differentiated mesenchymal cells (fibroblasts or myofibroblasts), known as the EMT, despite the fact that the EMT plays a significant role in embryonic development and tumorigenesis, there showed no evidence of the EMT in cell fate mapping studies in mice contributing to BDL-induced or CCl4-induced liver fibrosis. Since myofibroblasts are the main producer of ECM in fibrotic livers, they are a prime candidate for anti-fibrotic treatment. To improve the myofibroblast targeting, erlotinib, a small molecule epidermal growth factor receptor inhibitor, was loaded with platelet-derived growth factor receptor b (PDGFRB)-targeting peptide-nanoparticles (PPB-NP-erlotinib), and PPB-NP-erlotinib treatment exhibited a lower dose of erlotinib, and a greater effect in anti-fibrosis [14].

#### 2.1.5. Liver Sinusoidal Endothelial Cells

Liver sinusoidal endothelial cells are non-parenchymal cells, which form the sinusoidal wall and represent the interface between the blood and Kuffer cells on the one side and hepatocytes and HSCs on the other. In a healthy liver, LSECs are fully differentiated with high permeability. They are the most permeable endothelial cells in the mammalian body due to their association with fenestrae, absence of the diaphragm, and lack of the basement membrane. In human and rat fibrotic livers, LSECs lose their fenestrate and undergo capillarization. The capillarization of LSECs arises early in the course of NAFLD, probably as a result of LSEC exposure to dietary macronutrients. LSEC capillarization, in turn, encourages steatosis [15]. By keeping HSCs inactive through NO, healthy LSECs act as barriers against liver fibrosis, whereas changed LSECs (after capillarization and LSEC dysfunction) lose this capacity. Additionally, altered LSECs emit profibrogenic chemicals that stimulate HSCs, including TGF-b, Hedgehog molecules, laminin, and fibronectin [16]. Targeting LSECs to maintain the differentiation of LSECs is an interesting strategy in liver fibrosis therapy. Quantum dots (QDs) are used for imaging and transport of therapeutics [17]. A rapid absorption targets delivery of QDs with bound materials to the LSECs; 60% of an oral dose of QDs is rapidly distributed to the liver within 30 min, and this increases to 85% with FSA biopolymer coating. QDs coated with a biopolymer layer of FSA improve the bioavailability and delivery of metformin to LSECs.

### 2.2. Key Signaling Pathways in Liver Fibrosis

#### 2.2.1. TGF-b Signaling Pathway

The TGF-b family signaling is a core regulator in liver fibrosis, which induces fibrosis via canonical (Smad dependently) and non-canonical(non-Smad) pathways. As illustrated in Figure 2, TGF-b binds to single transmembrane type I and type II receptors that have serine-threonine kinase activity to signal its multifunctional actions on liver cells. In the canonical pathway, ligand (TGF-b) induces the formation of heteromeric complex, and the type II kinase trans-phosphorylates the type I receptor at specific sites, which then activates the type I kinase. Through the phosphorylation of particular proteins, including the receptor-regulated (R)-Smads, Smad2, and Smad3, intracellular signaling is initiated. The heteromeric complexes formed when activated Smad2 and Smad3 work with the common mediator Smad4 to translocate to the nucleus where they operate as transcription factors to control particular gene transcriptional responses (profibrotic genes) [18]. TGF-b also regulates additional signaling pathways via non-Smad signaling pathways, including those involving Wnt/ b-catenin, MAPK, mTOR, IKK, PI3K/Akt, and Rho GTPase, thereby resulting in liver fibrosis.

Due to the complication of the pathogenesis and mechanisms of liver fibrosis, effective targeting and drug delivery systems are required in the development of liver fibrosis therapeutics. In addition, to improve clinical efficiency, combination therapies impacting two or more targets are likely to be necessary.

The critical effects of the TGF-b signaling pathway in liver fibrosis make it promising as a therapeutic strategy. Thrombospondin 1 (TSP-1) is a matricellular protein that plays a role in regulation of latent TGF-b activation. The TSP-1 antagonist is able to mediate TGF-b activation in fibrotic diseases, thereby targeting the TSP-1/TGF-b pathway potentially in a selective way in liver fibrosis treatment [19]. Our lab reported that the extracellular matrix protein 1 (ECM1) plays a protective role in the development of liver fibrosis by keeping TGF-b in an inactive form via interaction with Integrin aV. The therapeutic efficacy of ectopic ECM1 expression in the chronic CCl4 liver fibrosis mouse model was tested, and the results indicated that ECM1 was successfully expressed in the liver upon AAV8 infection and partially rescued the CCl4 injury-mediated fibrosis [20]. The results strongly support the application of drugs that target the TGF-b signaling pathway as a promising strategy to treat liver fibrosis in patients.

#### 2.2.2. Wnt Signaling Pathway

The Wnt signaling pathway is one of the numerous intracellular signaling pathways that are involved in the pathophysiology of liver fibrosis, and its importance is increasing. The canonical (b-catenin-dependent) and non-canonical classes (b-catenin-dependent) pathways make up the majority of the Wnt signaling pathway; b-catenin acts as a protein with dual functions: adhesion molecule and transcription factor. In a healthy liver, b-catenin is located in the membrane of hepatocyte, while in the injured liver, it changes to the cytoplasm. In the inactivated state, b-catenin in the hepatocyte membrane forms a bridge between actin and E-cadherin. When Wnt signaling is off, b-catenin (in a multiprotein complex with GSK-3b, Axin, CK1a, b-TrCP, and APC) is phosphorylated by GSK-3b and CK1a and ubiquitinated by b-TrCP. In the end, the proteasome breaks down b-catenin. When Wnt signaling is active, Wnt-Fz and LRP work together to coordinate the activation of Dvl, which causes the multiprotein complex to dissociate and inactivate GSK-3b (no phosphorylation) [21]. Excessive free b-catenin translocates to the nucleus and binds to TCF/LEF transcription factors, resulting in transcriptional activation of Wnt target genes. Non-canonical that encompass the non-canonical Wnt/$Ca2^{+}$ pathway and PCP pathway is responsible for the activation of HSC through regulating proinflammatory cytokines and cellular proliferation and differentiation.

Numerous compounds, including antagonists, short interfering RNA (siRNA), soluble receptors, and the transcription inhibitors ICG-001 [22], and honokiol [23], have been found to suppress Wnt signaling. These compounds might be potential fibrosis therapeutic options.

#### 2.2.3. Hedgehog (Hh) Signaling Pathway

The Hedgehog (Hh) signaling pathway is another critical regulator in the progression of liver fibrosis. In many types of adult liver injury, the Hedgehog (Hh) signaling system, which regulates progenitor cell fate and tissue development throughout embryogenesis, occurs. The Hh pathway is silent in Hh-responsive cells when Hh ligands are absent. PTCH1 blocks Smo’s activity when the signaling is in the “Off state”. Protein kinases, such as PKA, CK1, and GSK3b phosphorylate Gli, cause proteolytic cleavage to produce Gli-repressor (Gli-R). The target genes’ expression is suppressed by Gli-R. In cells that secrete Hhn, the released Hhn then binds to PTCH1. Smo repression is eliminated as a result of PTCH1 degradation in endosomes, and the activated Smo prevents PKA from acting on Gli proteins, which causes SuFu to dissociate and Gli active form (Gli-A) to develop. Thereby, the target genes’ expression is encouraged by Gli-A [24]. There are numerous chemical Hedgehog inhibitors known, such as Gant61, GDC-0049, MD85, and vismodegib. Most of them have shown promise against liver fibrosis in vivo and in vitro. The cRGDyK-guided liposome loading vismodegib significantly reduced the fibrogenic phenotype in vivo. The modified vismodegib was delivered by the delivery system to active HSCs rather than quiescent HSCs, which resulted in preferential accumulation in the fibrotic liver. These results illustrate the promise of delivering therapeutic agents to activated HSCs to treat liver fibrosis [25].

## 3. Current Interventions in Liver Fibrosis Management

At present, the increasing understanding of common mechanisms and the identification of various mediators and pathways activated in liver fibrogenesis offer a multitude of targets for antifibrosis therapy. However, up to now, there are no FDA and EMA-approved drugs for the treatment of liver fibrosis. Antiviral therapy has made remarkable progress in preventing or even reversing the advancement of liver fibrosis, but such a strategy remains elusive for patients with advanced alcoholic steatosis, genetic, or autoimmune liver disease, especially nonalcoholic steatosis. Actually, the existing diagnosis methods limit the drug development of liver fibrosis to a certain extent. The diagnostic methods of liver fibrosis can be divided into three categories: liver biopsy, biomarkers, and imaging studies. Although liver biopsy has many limitations, including sampling bias and subjective scoring, it is currently the gold standard in clinical studies. At present, blood biomarkers used in clinical settings mainly include ALT, AST, LDL, HDL, and TG, which can be detected in hyperlipidemia. Emerging blood biomarkers include ELF, ProC3, and NIS4. ELF is currently used to assist in the diagnosis of patients with NASH with F3 and F4 fibrosis. Imaging tests mainly include ultrasound, fibroscan, and MRI(MRE, PDFF, cT1). Among these, ultrasound is commonly used in clinical practice. Fibroscan assesses the severity of liver fibrosis by measuring the degree of elasticity of the liver. The advantage of MRI (MRE, PDFF, cT1) over ultrasound and fibroscan is that the detection of the liver as a whole reduces the bias caused by sampling, but the price is relatively expensive. The conduct of clinical trials may require comprehensive diagnosis, and in any case, the development of non-invasive biomarkers is urgent for clinical applications [26]. Liver fibrosis therapy options include lifestyle modification, bariatric surgery, and pharmacologic treatments, which include small molecular inhibitors or agonists, proteins, antibodies, and natural compounds. Recently, new treatments for liver fibrosis have arisen, such as mesenchymal stem cells. A summary of medicines in clinical development is provided in Figure 3 and Table 1. In the next section, we will focus on different strategies and therapeutic targets proposed for liver fibrosis progression.

### 3.1. Pharmacologic Treatments

#### 3.1.1. Targeting Liver Lipid Metabolism and Oxidative Stress

##### Molecule Compounds

(1)ACC InhibitorsACC (acetyl CoA carboxylase) is a key enzyme regulating lipid metabolism. ACC comes in two informs, which present in different tissues: ACC1 is more highly expressed in liver and adipose tissue, whereas ACC2 is more expressed in oxidative tissues including cardiac muscle and skeletal muscle. Firsocostat (GS-0976), an ACC1 inhibitor, reduced de novo lipogenesis and liver fat in 126 patients with nonalcoholic steatohepatitis (NASH) without cirrhosis but historical biopsy consistent with NASH and F1–F3 fibrosis (NCT02856555). The administration of 20 mg GS-0976 for 12 weeks was shown to be safe, but there was a 13% increase in serum triglycerides, with hypertriglyceridemia being the most common AE (adverse event) [27].(2)SCD1 inhibitorsThe stearoyl-CoA desaturase 1 (SCD1) enzyme is a rate-limiting enzyme that regulates the monounsaturated fatty acid production process. The inhibition of SCD1 reduces fatty acid synthesis while increasing b-oxidation, resulting in lower hepatic triglycerides. Aramchol, a partial inhibitor of SCD1, forms a stable amide link between two natural components, cholic acid (bile acid) and arachidic acid (saturated fatty acid). NASH resolution without worsening fibrosis was obtained in 16.7% of Aramchol 600 mg against 5% of the placebo arm in a Phase IIb ARREST study NCT02279524, although the primary end point of a reduction in liver fat content with Aramchol 600 mg did not meet the expected significance level. The observed safety and changes in liver histology and enzymes provide a rationale for SCD1 modulation as a promising therapy for NASH and fibrosis and are currently being evaluated in a Phase 3 study [28].(3)FGF agonistsFibroblast growth factor functions as regulating bile acid synthesis and glucose homeostasis. NGM282 (aldafermin) is a FGF19 analogue. Patients were randomly assigned to receive 3 mg, 6 mg NGM282, or placebo in a study of 82 patients with NASH (NCT02443116) [29]. The absolute change in liver fat content from baseline to week 12 was the primary outcome. The 12 weeks of NGM282 treatment resulted in relative decreases in corrected T1 (cT1; 8% and 9%), ALT (67% and 60%), AST (57% and 52%), and Pro-C3 (22% and 33%). Aldafermin-related AEs were few, mild to moderate, and gastrointestinal in origin [30].Pegbelfermin(BMS-986936) is a pegylated FGF21 analog. The safety, pharmacokinetics, and pharmacodynamic effects of BMS-986036 in 184 people with NASH were evaluated in a 16-week randomized, double-blind trial (NCT02413372) [31]. The group receiving 10 mg pegbelfermin daily (68% vs. 13%; *p* = 0.0004) and the group receiving 20 mg pegbelfermin weekly (52% vs. 13%; *p* = 0.008) showed a substantial decrease in absolute hepatic fat fraction compared with the placebo group at week 16. Most AEs were mild [32].(4)THR beta agonistsThyroid hormone acts as a ligand for two receptors, thyroid hormone receptor-a (THRa) and thyroid hormone receptor-b (THRb). THRa is highly expressed in the heart and bone, whereas THRb is the primary form expressed in the liver, which regulates several processes involved in hepatic triglyceride and cholesterol metabolism to lower serum cholesterol and intrahepatic lipid content [33]. Low thyroid function was associated with a higher prevalence of NASH and advanced fibrosis than strict-normal thyroid function [34].Resmetirom and VK2809 are two orally active THR agonists that are liver-directed and have a several-fold higher selectivity for THRb than THRa. A Phase 2, randomized, double-blind, placebo-controlled study evaluated the effect of MGL-3196 on patients with NASH (fibrosis Stages 1–3) [35]. Patients receiving resmetirom showed a relative reduction of hepatic fat compared with placebo at week 36 (−37.3% versus −8.5; *p* < 0.0001). AEs were largely mild or moderate and were evenly distributed between groups [36]. Further studies of resmetirom will allow assessment of the safety and effectiveness of resmetirom in a larger number of patients with NASH with the possibility of documenting associations between histological effects and changes in imaging. Resmetirom has a larger total fat-burning effect than the other targets we discussed above, and the most recent Phase 3 results show a positive result in NAFLD. So, it is promising to be the first FDA-approved treatment for NASH. One of the disadvantages of resmetirom is that the metabolized product is a stable and active product, which means that the drug must be adjusted twice to account for individual metabolic differences, which might be inconvenient to use therapeutically.(5)FXR agonistsThe nuclear receptor FXR (farnesoid X receptor) is mostly expressed in the liver, gallbladder, and intestine. Bile acids, which regulate lipid metabolism and glucose homeostasis, increase insulin sensitivity, and have anti-inflammatory and anti-fibrosis properties, are the natural ligands of FXR [37].Obeticholic acid (OCA) is the most advanced drug in FXR agonists. In a Phase III REGENERATE trial (NCT02548351), 1968 patients with definite NASH were randomly assigned to receive OCA at 10 mg or 25 mg daily or placebo [38]. After 18 months of treatment, the fibrosis improvement end point was achieved by 23% in the OCA 25 mg group (*p* = 0.0002) versus 12% of patients in the placebo group. Pruritus was the most common AE. Patients with OCA therapy were associated with an increase at 12 weeks in small very low-density lipoprotein (VLDL) and LDL and a reduction in high-density lipoprotein (HDL) [39]. In conclusion, the role of OCA in anti-fibrosis was present, but its overall effectiveness was only about 25%, suggesting that only one out of every four people responds [40]. At the same time, the AEs were rather strong, including 50% pruritus and an increase in LDL, both of which are not good for people with cardiovascular disease [41]. Intercept will resubmit the NDC with 500 additional samples, and the patients in the group have also been observed for a longer period of time, although the specific outcomes have yet to be revealed.Cilofexor (GS-9674) is a selective nonsteroidal FXR agonist that showed promise in a Phase II trial in NASH patients. In this double-blind, placebo-controlled trial, 140 patients with NASH without cirrhosis were randomized to receive cilofexor 100 mg, 30 mg, or placebo orally once daily for 24 weeks (NCT02854605) [42]. Cilofexor was generally well tolerated. Moderate to severe pruritus was more common in patients receiving cilofexor 100 mg (14%) than in those receiving cilofexor 30 mg (4%) and placebo (4%).(6)PPAR agonistsPeroxisomal proliferator-activated receptors (PPARs) are nuclear receptors that regulate metabolic signaling pathways, cell differentiation, and immune inflammation [43]. There are three different patterns of tissue expression and functional activity. PPARa is expressed mostly in active metabolism tissues such as the liver, kidney, heart, and skeletal tissue. It has been demonstrated to lower triglycerides and enhance HDL-C in serum by regulating lipid delivery through fatty acid transport and b-oxidation [44]. PPARg regulates lipogenesis, fatty acid differentiation, and glucose homeostasis. As a sensor of insulin, PPARg reduces ectopic fat accumulation by increasing the storage of fatty acids such as TG. It can also reduce inflammation by inhibiting the production of cytokines [45]. PPARd has a ubiquitous tissue expression profile, and its function is involved in fatty acid metabolism [46].Elafibranor is a dual agonist for PPARa and PPARd that improves insulin sensitivity, glucose homeostasis, and lipid metabolism. In double-blind, placebo-controlled research, patients with NASH without cirrhosis were randomly assigned to one of three groups: elafibranor 80 mg, 120 mg, or placebo daily for 52 weeks (NCT01694849) [47]. NASH resolution was reduced with elafibranor 120 mg daily, without fibrosis worsening. It was well tolerated and did not cause weight gain. Subsequently, the effect of elafibranor 200 mg daily for 72 weeks was assessed in a Phase III trial (NCT02704403). The interim analysis, however, found no evidence of a therapeutic effect, with a response rate of 19.2% in the elafibranor arm against 14.7% in the placebo arm in NASH resolution without worsening fibrosis.(7)GLP-1 receptor agonistsGlucagon-like peptide-1 (GLP-1) is a gut-derived incretin hormone that induces weight loss and insulin sensitivity by improving insulin synthesis and release, reduces glucagon secretion and hepatic gluconeogenesis, reduces steatosis, and affects lipid metabolism by decreasing lipogenesis and increasing oxidation to metabolize fatty acids [48]. In mouse models of NASH, GLP-1 analogs have been demonstrated to lower liver enzymes, oxidative stress, and liver histopathology. Endogenous GLP-1 is degraded within minutes in vivo by the enzyme dipeptidyl peptidase-4 (DP-4), whereas liraglutide is a long-term GLP-1 receptor agonist that has been licensed for the treatment of T2DM [49].Patients with overweight and NASH were investigated in a 48-week, randomized, double-blind, placebo-controlled Phase II experiment (NCT01237119) [50]. The primary outcome resolution of definite NASH with no worsening in fibrosis was met in patients receiving liraglutide. Patients receiving liraglutide showed a significantly mean BMI decrease versus the placebo group from baseline. Unfortunately, there was no statistically significant difference in histologic response between the liraglutide and placebo groups (including hepatocyte ballooning score, steatosis, lobular inflammation, Kleiner fibrosis stage, and total NAFLD activity score), indicating that the benefits of liraglutide were possibly related to weight loss rather than liraglutide treatment. The majority of the AEs were mild to moderate in severity.A 72-week, double-blinded Phase II trial involving patients with NASH and Stage F1-3 fibrosis as confirmed by biopsy, was evaluated to compare semaglutide with placebo (NCT02970942) [51]. The primary end point of NASH remission and no worsening of fibrosis was fulfilled for all doses of semaglutide compared to placebo [52]. However, a recent study found that semaglutide failed to cure NASH with F4 fibrosis, indicating that it is difficult to reverse advanced fibrosisonly metabolic homeostasis and that combined therapy is required for advanced fibrosis and/or cirrhosis treatment.

##### Antibodies

A full-length, humanized antiFGFR1c/KLB agonist antibody, BFKB8488A, selectively activates FGFR1 in a KLB-dependent manner and imitates the effects of FGF21. The safety, tolerability, and pharmacokinetic (PK) effects of BFKB8488A were evaluated in a Phase 1b trial (NCT03060538), and the results showed that BFKB8488A was safe and well tolerated in patients with both T2D and NAFLD, with nausea being the most notable side effect. A Phase 2b clinical trial was conducted to evaluate the efficacy and safety of NASH patients with Stage 2–3 fibrosis (NCT04171765, recruiting) [53].

Another monoclonal antibody, NGM313, selectively activates FGFR1c/KLB by directly binding to KLB and has been demonstrated to significantly reduce liver fat in patients with NAFLD (NCT03298464). The effect of MK-3655 (rename) will be evaluated in individuals with pre-cirrhotic NASH (NCT04583423, recruiting). We may witness the first-approved antibodies for the treatment of NASH if these mAbs succeed.

#### 3.1.2. Targeting Liver Inflammation and Cell Death

##### Antibodies

(1)CB1 antibodyCannabinoid receptor 1 (CB1R) is a G-protein coupled receptor, whose expression is unregulated in the liver with viral hepatitis, alcoholic and nonalcoholic fatty liver disease, and cirrhosis. The inhibition of CB1R exhibited a beneficial effect in improving liver function by suppressing the activation of HSCs [54]. Nimacimab, also known as RYI-018, is an antagonist antibody, and in a multicenter, adaptive design, randomized, parallel-group Phase I trial, the safety, tolerability, and PK of repeat IV doses of RYI-018 in participants with NAFLD were examined (NCT03261739). In addition, a Phase IIb clinical trial is going to be conducted.(2)Anti-CCL24 AntibodyCCL24 is a chemokine that promotes immune cell trafficking and activation as well as profibrotic activities through the CCRs receptor. A blocking mAb that targets CCL4 and CM-101 was evaluated in a Phase 1b trial, and the results showed that CM-101 treatment decreased fibrosis and inflammatory biomarkers in serum without compromising safety or tolerability [55]. The effect and safety of CM-101 will be evaluated in NASH patients with Stage 2–3 fibrosis.

##### Natural Compounds

(1)Berberine (BBR)Along with chemical compounds, various natural compounds have interestingly been prescribed for the treatment of liver fibrosis. Natural compounds with a diverse structure, low toxicity, and a wide range of sources have snatched up a lot of attention in recent years for their multiple-target actions in liver disease.Berberine (BBR) is a benzylisoquinoline alkaloid with various pharmacological activities, such as antiparkinson, anti-cancer, anti-diabetic, anti-obesity, anti-inflammatory, anti-viral, and cardiovascular actions. These potential biological activities of BBR were attributed due to their ability to interact with various biological targets involved in the pathogenesis of various diseases, including liver fibrosis. Based on the efficacy of HTD1801 (berberine ursodeoxycholate, ionic salt of berberine, and ursodeoxycholic acid) in animals, a Phase II randomized controlled trial of two doses of HTD1801 versus placebo in 100 patients with fatty liver disease and diabetes was conducted in 100 patients (NCT03656744) [56]; 18 weeks treatment of HTD1801 (1000 mg twice a day) reduced liver fat content assessed by MRI-PDFF (mean absolute decrease −4.8% in HTD1801 arm versus −2.0% in the placebo arm; *p* = 0.011) with gastrointestinal AEs.(2)SilymarinSilymarin is a lipophilic complex of two flavonoids (quercetin and taxifolin) and three flavonolignane diastereomers: silibinin, silydianin, and silychristin. A randomized, double-blind, placebo-controlled trial (NCT02006498) was conducted to assess the effect of silymarin on patients with NASH and NAFLD activity scores (NAS) of at least four in patients with NASH and NAFLD (proved by biopsy) [57]. Patients who received silymarin (700 mg daily for 48 weeks) did not have their NAS scores reduced by 30% compared to those who received a placebo. The therapeutic impact of silymarin in the treatment of NAFLD was investigated in a meta-analysis (PRISMA) of randomized control trials involving 587 people with NAFLD [58]. Silymarin was found to dramatically lower AST and ALT levels when compared to the control group, implying that it could be useful phytotherapy for NAFLD patients [59].

#### 3.1.3. Targeting Liver Fibrosis

##### Antibodies

Endoglin (CD105) is a membrane receptor found on endothelial cells that are highly expressed in the proliferating tumor vasculature. The overexpression of CD105 drastically enhanced the activation of Smad1/5/8, and the phosphorylation of ERK1/2 was increased, which indicated that suppression of CD105 may play a role in the development of liver fibrosis [60]. TRC105 is a chimeric IgG1 CD105 monoclonal antibody that inhibits angiogenesis and causes proliferating endothelial apoptosis and antibody-dependent cellular cytotoxicity (ADCC).

##### Gene Therapy

Heat shock protein (HSP47) is a collagen-binding molecular chaperone located in the endoplasmic reticulum that is required for collagen synthesis and deposition, both of which are important aspects of fibrogenesis. In mice models lacking HSP47, the advancement of liver fibrosis is dramatically slowed [61]. In a Phase 1 trial on healthy persons (n = 20), ND-LO2-s0201, a lipid nanoparticle that targets hepatic stellate cells and is capable of reversibly suppressing HSP47, was shown to be well tolerated and may be investigated further in Phase II trials. A randomized, double-blind, placebo-controlled Phase II study evaluated the safety and efficacy of BMS-986263 in patients with advanced hepatic fibrosis after virologic cure of hepatitis C (an infection caused by a virus that attacks the liver and leads to inflammation, NCT03420768) [62]. BMS-986263 is a lipid nanoparticle delivering siRNA designed to degrade HSP47 mRNA. Patients who took BMS-986263 for 36 weeks resulted in METAVIR and Ishak score improvements. All AEs were mild or moderate in intensity. Further evaluation of BMS-986263 in patients with active fibrogenesis is warranted.

##### Cell Therapy

The liver function of advanced cirrhosis or end-stage liver disease is irreversibly impaired, and the only curative treatment is orthotropic liver transplantation (OLT). Due to the significant drawbacks of OLT, it is important to investigate alternative therapeutic approaches. The first experimental transplant of primary human hepatocytes (PHHs) opened up the possibility of cell treatment in 1976. Since then, rapid development has taken place in cell treatment for chronic liver illnesses. The two main mechanisms by which donor stem or progenitor cells function are differentiation into functional cells to replace damaged cells, and production of bioactive factors to promote the proliferation or maturation of the patient’s own tissue-resident progenitor cells and immunoregulatory factors to control the progression of inflammation [63].

There are numerous cell types that can be used in the treatment of liver fibrosis and cirrhosis, such as mesenchymal stem cells (MSCs), umbilical cord mesenchymal stem cells (UCMSCs), bone marrow mesenchymal stem cells (BMMSCs), liver-derived mesenchymal stem cells (LDMSCs), and so on. Stem cell-based therapies are less intrusive for patients than surgical procedures and have a minimal risk of immunological rejection when compared to traditional therapeutic approaches [64]. In a CCl4-induced liver cirrhosis mouse model, hair follicle mesenchymal stem cells (HF-MSCs) with ECM1 modification exhibited a better effect on promoting injured liver repair and improving liver function, which may provide a relevant theoretical basis for the application of ECM1-HF-MSCs in liver cirrhosis treatment [65].

#### 3.1.4. Combination Drug Trials

Combination therapy impacting two or more targets is likely to be required due to the complexity of etiology. Furthermore, the response rate of most single-drugs is too low; for instance, OCA’s response rate is just 25%, and most single-drug responses do not exceed 30%. The following concepts of combination medicines were shown to achieve improved efficacy and reduce related AEs of single drugs: (1) drugs that target several pathways; (2) medicines that target diverse characterizations of the disease; (3) drugs that combine small-molecule and macromolecular drugs; and (4) drugs that target metabolism and liver fibrosis. Combination therapies currently available can be divided into the following categories:Mechanistically complementary, such as the combination of an ACC1/2 inhibitor (PF-05221304) and a DGAT2 inhibitor (PF-06865571): In participants with NAFLD, a Phase II trial (NCT03776175) was conducted to investigate the effects of PF-05221304 alone, PF-06865571 alone, and the combination of PF-05221304 and PF-06865571 [66]. The results indicated that PF-05221304 monotherapy exhibits a great anti-steatosis effect; however, elevation in serum TG with a high dose of PF-05221304 limits the utility for those with cardio-metabolic diseases. Co-administration of PF-05221304 with PF-06865571 may be a viable approach for overcoming the limitations of PF-05221304 monotherapy and achieving better therapeutic benefits.Targeting different disease characteristics, such as FXR (targeting fibrosis and inflammation) and ACC (reducing fat accumulation): A combination of the ACC inhibitor GS-0976 (firsocostat) and the nonsteroidal FXR agonist GS-9674 (cilofexor) trial was observed to improve hepatic steatosis, biochemistry, and stiffness in patients with NASH (NCT02781584) [67]. However, no clinically significant reductions in total LDL or HDL were observed with any of the programs.Combination of small-molecule drugs tropifexor (LJN452) and cenicriviroc (CVC): A randomized, double-blind study was conducted to evaluate the safety, tolerability, and efficacy of a tropifexor (LJN452) and cenicriviroc (CVC) combination treatment in patients with NASH and liver fibrosis (NCT03517540) [68]. In addition, the results suggested that when compared to monotherapy, combination therapy is likely to provide extra benefits.Small-molecule drug medications combined with macromolecular drugs: In patients with NASH, the FXR agonist cilofexor (GS-9674), the acetyl-coenzyme A carboxylase inhibitor firsocostat (GS-0976), and the GLP1 receptor agonist semaglutide were evaluated (NCT03987074) [69]. Semaglutide plus firsocostat and/or cilofexor was generally well tolerated in patients with mild-to-moderate fibrosis owing to NASH. Treatment resulted in additional improvements in liver steatosis and biochemistry compared to semaglutide alone in exploratory efficacy studies.Dual-target macromolecular drugs: LY3298176 is a novel dual glucose-dependent insulinotropic polypeptide (GIP) and glucagon-like peptide-1 (GLP-1) receptor agonist that has been developed for the treatment of type 2 diabetes [70]. GLP1 receptor agonists and dual GLP1-GIP agonists have been reported to be associated with significant weight loss approaching 10% or more. In a Phase II randomized, double-blind trial (NCT03131687), LY3298176 was found to have significantly greater efficacy than GLP1R agonist in terms of glucose management and weight loss, with an acceptable safety and tolerability profile. According to the most recent data, individuals taking trizepatide 15 mg once a week lost a significant amount of weight (−22.5%), demonstrating its efficacy in the treatment of NASH with fatty liver. At 26 weeks, the 10 mg and 15 mg tirzmepatide groups showed significant reductions in ALT, AST, K-18, and Pro-C3, whereas adiponectin increased significantly. In order to compare the safety and efficacy of tirzmepatide vs. placebo in individuals with NASH, 196 people were recruited (SYNERGY-NASH, NCT04166773). Mounjaro (tirzepatide), developed by Eli Lilly, has been granted FDA and EMAs approval for use in combination with diet and exercise to improve glycemic control in adults with type 2 diabetes.

**Table 1 ijms-23-12572-t001:** Current clinical trials in liver fibrosis.

Target	Categories	Agent	Phase of Clinical Trial	Ref.
Targeting liver lipid metabolism and oxidative stress	Molecule compounds	ACC inhibitors Firsocostat PF-05221304	Phase II Phase II	[27,70,71]
		SCD1 inhibitors Aramchol	Phase III	[28]
		FGF agonists Aldafermin Pegbelfermin Efruxifermin	Phase II Phase II Phase II	[29,30,31,32]
		THR beta agonists Resmetirom K2809-201	Phase III Phase II	[35,36,37]
		FXR agonists Obeticholic acid Tropifexor Cilofexor	Phase III Phase II Phase II	[38,39,40,42,72]
		PPAR agonists Elafibranor Lanifibranor Saroglitazar Seladelpar	Phase III Phase II Phase II Phase II	[47,73,74,75]
		ASK1 inhibitors Selonsertib	Phase III	[76]
		GLP-1 receptor agonists Liraglutide Semaglutide Trizepatide	Phase II Phase II Phase II	[50,51,52,77]
		SGLT inhibitors Ipragliflozin Dapagliglozin	Phase II Phase II	[78]
	Antibodies	FGFR1c/KLB	Phase II	[53]
Targeting liver inflammation and cell death	Molecule compounds	CCR2/CCR5 inhibitors Cenicriviroc	Phase III	[79]
		Mitochondrial pyruvate carrier MSDC-0602K	Phase II	[80]
	Antibodies	CB1 antibody Nimacimab	Phase I	[54]
		Anti-IL-17 antibody Secukinumab	Phase I	[81]
		Anti-CCL24 antibody CM-101	Phase II	[55]
	Natural compounds	Berberine (BBR)	Phase II	[56]
		Silymarin	Phase II	[57]
		Resveratrol (RES)	Phase II	[82]
Targeting liver fibrosis	Molecule compounds	Galectin 3 inhibitors GR-MD-02	Phase III	[83]
		ARB inhibitors Losartan	Phase III	[84,85]
		Tyrosine kinase inhibitors Sorafenib	Phase II	[86,87]
	Antibodies	LOXL2 inhibitors	Phase II	[88]
		Integrin aV inhibitors 3G9	Phase II	[89]
		CD105 inhibitors TRC105	Phase II	[60]
	Gene therapy	siHSP47	Phase II	[62]

## 4. Conclusions and Future Perspectives

The health burden of liver fibrosis is increasing worldwide, which is initiated by chronic liver disease, and eventually cirrhosis and HCC. Each hepatic and recruited cell type contributes to liver fibrosis. Although there are no FDA and EMA-approved drugs for liver fibrosis treatment, multiple promising anti-fibrosis agents have successfully reversed liver fibrosis in clinical trials and experimental models. Understanding the pathogenesis and mechanisms of liver fibrosis is important to developing strategies to deal with the disease. Both hepatic cells and other organs, such as the intestine and adipose tissues, influence the progression of liver fibrosis, although most research focuses on HSC and MF. In addition to targeting the liver, more and more drugs target other organs in clinical trials, such as intestine (GLP-1 agonists) and kidney (SGLT2 inhibitors). Icaritin is the first natural product-derived drug for the treatment of HCC, and it has been approved for patients with unresectable HCC in China, which indicates the vital role of natural compounds in liver fibrosis therapies. In more complicated liver diseases, NAFLD is associated with genetic risk and diabetes, and lifestyle modification supported by pharmacological treatment might be necessary. Although certain anti-fibrotic agents have shown to attenuate the progress of liver fibrosis, there are still significant obstacles. Firstly, early identification of patients with liver fibrosis is needed to improve patient outcomes. Liver biopsy assessment is the gold standard at present, as the optimal strategy for the advanced liver disease remains to be established. Especially, non-invasive imaging- and biomarker-based diagnosis are required for assessing the progression of liver fibrosis and for easy observation for the outcomes of clinical trials. Secondly, a crucial point to be taken into account is the pro-fibrogenic mechanisms. The long-term activation of HSCs is an important aspect to be considered when addressing the development of liver fibrosis. Moreover, apoptosis and inflammation are believed responsible for liver fibrosis progression. Thirdly, the main barrier is drug delivery to target cells, and nanocarriers show great potential for specific drug delivery to target cells. Apart from physical entrapment into NPs or micelles, another way to improve drug loading and specific delivery is conjugated with selective receptors that are expressed on target cells. Lastly, a suitable experimental model (in vitro and in vivo) should be chosen to mimic the development of liver fibrosis for the efficient identification of anti-fibrogenic drugs, which appropriately exhibit the cross talk between pro-fibrogenic cells and immune cells.

## Figures and Tables

**Figure 1 ijms-23-12572-f001:**
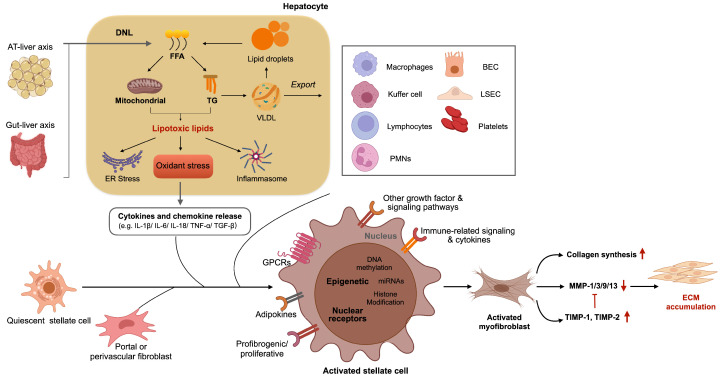
Molecules and mechanisms of liver fibrogenesis.

**Figure 2 ijms-23-12572-f002:**
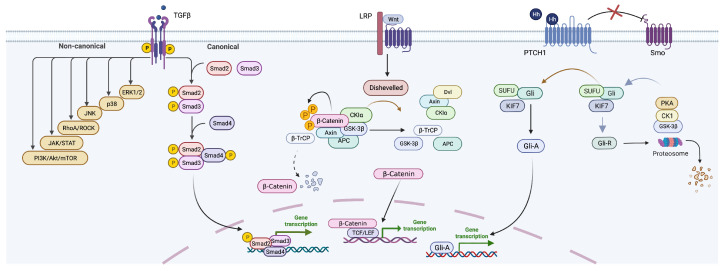
Key signaling pathways in liver fibrogenesis.

**Figure 3 ijms-23-12572-f003:**
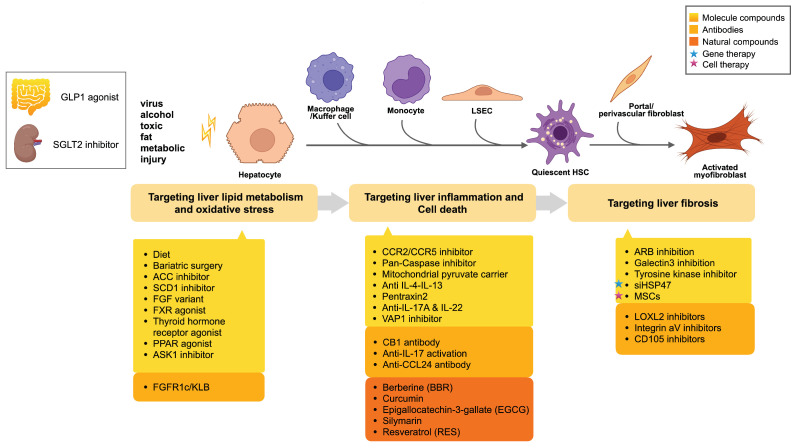
Therapeutic opportunities for blocking fibrosis development.

## Data Availability

All data are contained within the article.

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
