# Peer review of "The Molecular Mechanisms of Liver Fibrosis and Its Potential Therapy in Application"

_ijms, 2022, doi:10.3390/ijms232012572_

Round 1

Reviewer 1 Report

The article by Zhang D, Zhang Y and Sun B entitled “The molecular mechanisms of liver fibrosis and its potential therapy in application” touches a very interesting hepatological topic. Alcoholic or non-alcoholic fatty liver disease is today the main chronic liver disease worldwide, with potential progression to cirrhosis and hepatocellular carcinoma. It is also well known that the prognosis of patients with fatty liver depends on the degree of liver fibrosis, a fact that makes a review on the biological mechanisms of fibrosis and related drugs, extremely interesting. The authors handle this issue successfully. They provide an up-to-date picture of the complex biological mechanisms of fibrosis and adequately explain the pathophysiological mechanisms underlying the action of the relevant drugs under investigation. The article provides good information to those interested on the subject in question. The text needs some changes, as noted in my comments that follow.

General Comments:

1.     Too many abbreviations. Consider reduction. An uninitiated reader will have difficulties in reading and comprehending the text...

2.     English language problems. Text needs careful re-reading and language corrections. Minor typing errors.

Major Comments:

1.       (Line 210-211): Please, add reference.

2.       (Line 372): For how long-time treatment lasted?

3.       (Lines 401-406): Please mention the current status of elafibranor.

4.       (Lines 4020-424): There is an apparent contradiction between sentences at lines 420,421 vs. 423, 424.

5.       (Line 492): Please add the reference by Kalopitas G, et al. (Nutrition 2021 Mar;83:111092. doi: 10.1016/j.nut.2020.111092. Epub 2020 Nov 25. Impact of Silymarin in individuals with nonalcoholic fatty liver disease: A systematic review and meta-analysis) and note that the authors call attention to potential flaws of the quality of the included studies.

6.       (Line 589): Authors may add that tirzapetide (Mounjano, Eli Lilly) has been granted FDA & EMAs approval for treatment of type 2 diabetes (2022).

Minor Comments:

1.     (Line 7): Please consider the change: “...with advanced viral, alcoholic or nonalcoholic steatosis...”

2.     (Line 9): Please omit the word "advanced".

3.     (Line 9): Please replace "followed" with "associated with".

4.     (Line 57-61): Consider changing or omitting the sentence.

5.      (Line 81-82) Consider the change: “…which is mostly fueled through absorption of dietary carbohydrates in the intestine…”.

6.     (Line 92-97): This sentence appears to be out of context. It may be omitted.

7.     (Line 128-131): The authors must clarify the meaning of this sentence.

8.     (Line 135): What "other HSC's" are you referring to?

9.     (Line 185): Please consider modifying the phrase to a more intelligible form.

10. (Lines 201-202): Unintelligible phrase. Consider rephrasing.

11.  (Lines 244-245): “Additionally, ECM1 is downregulated in liver fibrosis, which made it a potential biomarker in clinical.”: Incomplete sentence. Consider change.

12. (Line 390): Please explain what is the meaning of the “uncle receptors”.

13. (Line 440): “NTC04171765, recruiting".

14. (Line 470): "benzylisoquinoline".

Author Response

Dear reviewer, thank you for your appreciation and provided insightful comments about our manuscript. It’s our pleasure to share this review with you, please see below, in red, for a point-by-point response to your comments and concerns.

Comments and Suggestions for Authors

The article by Zhang D, Zhang Y and Sun B entitled “The molecular mechanisms of liver fibrosis and its potential therapy in application” touches a very interesting hepatological topic. Alcoholic or non-alcoholic fatty liver disease is today the main chronic liver disease worldwide, with potential progression to cirrhosis and hepatocellular carcinoma. It is also well known that the prognosis of patients with fatty liver depends on the degree of liver fibrosis, a fact that makes a review on the biological mechanisms of fibrosis and related drugs, extremely interesting. The authors handle this issue successfully. They provide an up-to-date picture of the complex biological mechanisms of fibrosis and adequately explain the pathophysiological mechanisms underlying the action of the relevant drugs under investigation. The article provides good information to those interested on the subject in question. The text needs some changes, as noted in my comments that follow.

General Comments:

  1. Too many abbreviations. Consider reduction. An uninitiated reader will have difficulties in reading and comprehending the text...
  2. English language problems. Text needs careful re-reading and language corrections. Minor typing errors.

A: Dear reviewer, thank you for your appreciation and provided insightful comments about our manuscript. It’s our pleasure to share this review with you, please see below, in red, for a point-by-point response to the reviewers’ comments and concerns.

  • 1) & 2) As you mentioned above, because the mechanisms and clinical trials of liver fibrosis are complicated and diversified, and the review covers a wide range of content, from mechanisms to clinical research and possible drug delivery systems, in which we use a lot of abbreviations, and we have reduced most of the abbreviations and made language corrections according to your suggestion.

Major Comments:

  1. (Line 210-211): Please, add reference.

A: Thank you for pointing this out, we have added the related reference.

  1. (Line 372): For how long-time treatment lasted?

A: As the data are shown in the paper “Obeticholic acid for the treatment of non-alcoholic steatohepatitis: interim analysis from a multicentre, randomised, placebo-controlled phase 3 trial”,  between Dec 9, 2015, and Oct 26, 2018, 1968 patients with stage F1–F3 fibrosis were enrolled and received at least one dose of study treatment; 931 patients with stage F2–F3 fibrosis were included in the primary analysis (311 in the placebo group, 312 in the obeticholic acid 10 mg group, and 308 in the obeticholic acid 25 mg group). The fibrosis improvement endpoint was achieved by 37 (12%) patients in the placebo group, 55 (18%) in the obeticholic acid 10 mg group (p=0·045), and 71 (23%) in the obeticholic acid 25 mg group (p=0·0002).  And the primary endpoints for the month-18 interim analysis were fibrosis improvement (≥1 stage) with no worsening of NASH, or NASH resolution with no worsening of fibrosis, with the study considered successful if either primary endpoint was met. 

    We have rephrased it to “After 18 months of treatment, the fibrosis improvement endpoint was achieved by 23% in the OCA 25 mg group (p=0.0002) versus 12% of patients in the placebo group.”

  1. (Lines 401-406): Please mention the current status of elafibranor.

A: Thank you for pointing this out, we have added “Subsequently, the effect of elafibranor 200 mg daily for 72 weeks was assessed in a phase III trial (NCT02704403). The interim analysis, however, found no evidence of a therapeutic effect, with a response rate of 19.2% in the elafibranor arm against 14.7% in the placebo arm in NASH resolution without worsening fibrosis.”

  1. (Lines 4020-424): There is an apparent contradiction between sentences at lines 420,421 vs. 423, 424.

A: Thank you for pointing this out, as illustrated in “Liraglutide safety and efficacy in patients with non-alcoholic steatohepatitis (LEAN): a multicentre, double-blind, randomised, placebo-controlled phase 2 study.” The primary outcome resolution of definite NASH with no worsening in fibrosis was met in patients receiving liraglutide (39% in the liraglutide arm versus 9% in the placebo arm; p=0.019).  In addition, 9% of liraglutide group versus 36% of placebo group had progression of fibrosis (p=0.04). Patients receiving liraglutide showed a significantly mean BMI decrease versus the placebo group from baseline. Unfortunately, there was no statistically significant difference in histologic response (including hepatocyte ballooning score, steatosis, lobular inflammation, Kleiner fibrosis stage, and total NAFLD activity score) between the liraglutide and placebo groups, indicating that the benefits of liraglutide were possibly related to the weight loss rather than Liraglutide treatment. Patients with cirrhosis were included in this trial to emphasize the safety of liraglutide, and their inclusion did not exaggerate the histologic responses, indicating that liraglutide is effective in the treatment of advanced fibrosis and/or cirrhosis. The majority of the AEs were mild to moderate in severity. In summary, Liraglutide has a certain improvement in liver function, but this effect is not statistically significant. was found to be safe, well-tolerated, and resulted in the histological clearance of NASH in the liver. Nevertheless, due to the small sample size, this study had some intrinsic limitations (26 patients assigned to each treatment group).

    We have rephrased it to “Patients receiving liraglutide showed a significantly mean BMI decrease versus the placebo group from baseline. Unfortunately, there was no statistically significant difference in histologic response between the liraglutide and placebo groups (including hepatocyte ballooning score, steatosis, lobular inflammation, Kleiner fibrosis stage, and total NAFLD activity score), indicating that the benefits of liraglutide were possibly related to weight loss rather than liraglutide treatment. The majority of the AEs were mild to moderate in severity.”

  1. (Line 492): Please add the reference by Kalopitas G, et al. (Nutrition 2021 Mar;83:111092. doi: 10.1016/j.nut.2020.111092. Epub 2020 Nov 25. Impact of Silymarin in individuals with nonalcoholic fatty liver disease: A systematic review and meta-analysis) and note that the authors call attention to potential flaws of the quality of the included studies.

A: Thank you for pointing this out, we have added the related reference.

  1. (Line 589): Authors may add that tirzapetide (Mounjano, Eli Lilly) has been granted FDA & EMAs approval for treatment of type 2 diabetes (2022).

A: Thank you for pointing this out, we have rephrased it to “According to the most recent data, individuals taking trizepatide 15 mg once a week lost a significant amount of weight (-22.5%), demonstrating its efficacy in the treatment of NASH with fatty liver. At 26 weeks, the 10 mg and 15 mg tirzmepatide groups showed significant reductions in ALT, AST, K-18, and Pro-C3, whereas adiponectin increased significantly. In order to compare the safety and efficacy of tirzmepatide vs placebo in individuals with NASH, 196 people are recruited(SYNERGY-NASH, NCT04166773). Mounjaro (tirzepatide), developed by Eli Lilly, has been granted FDA and EMAs approval for use in combination with diet and exercise to improve glycemic control in adults with type 2 diabetes.”

Minor Comments:

  1. (Line 7): Please consider the change: “...with advanced viral, alcoholic or nonalcoholic steatosis...”

A: Thank you for pointing this out. We have made corresponding corrections according to your suggestion.

  1. (Line 9): Please omit the word "advanced".

A: Thank you for pointing this out. We have made corresponding corrections according to your suggestion.

  1. (Line 9): Please replace "followed" with "associated with".

A: Thank you for pointing this out. We have made corresponding corrections according to your suggestion.

  1. (Line 57-61): Consider changing or omitting the sentence.

A: Thank you for pointing this out. We have made corresponding corrections according to your suggestion.

  1. (Line 81-82) Consider the change: “…which is mostly fueled through absorption of dietary carbohydrates in the intestine…”.

A: Thank you for pointing this out. We have made corresponding corrections according to your suggestion.

  1. (Line 92-97): This sentence appears to be out of context. It may be omitted.

A: Thank you for pointing this out. We have made corresponding corrections according to your suggestion.

  1. (Line 128-131): The authors must clarify the meaning of this sentence.

A: Thank you for pointing this out. Sorry for inserting the wrong example.

    We have corrected it to “The promotion of macrophage from pro-regenerative M2 to inflammatory M1, which led to hepatic injury induction and liver restoration improvement. The expression of SYK was induced in M1 macrophage, the specific inhibitor of SYK pathway can significantly reduce inflammation. a PLGA nanoparticles-based delivery of SYK pathway inhibitor (R406), exhibited stronger ameliorated steatosis, inflammation, and fibrosis. ”

  1. (Line 135): What "other HSC's" are you referring to?

A: Thank you for pointing this out. As illustrated in “Molecular and cellular mechanisms of liver fibrosis and its regression,” quiescent hepatic stellate cells (HSCs) are located in the space of Disse, a designated area between sinusoidal endothelial cells (LSEC) and hepatocytes clusters.

    We have rephrased it to “HSCs are found in the subendothelial area of the Disse and have close connections with the surrounding hepatocytes and sinusoidal endothelial cells (SECs).”

  1. (Line 185): Please consider modifying the phrase to a more intelligible form.

A: Thank you for pointing this out. As illustrated in “Liver fibrosis: Direct antifibrotic agents and targeted therapies,”In short bouts of acute liver disease fibrogenesis is matched by an upregulated fibrolysis (removal of excess ECM by proteolytic enzymes), mainly via the ECM degrading matrix metalloproteinases (MMPs), such as MMP-1, −3, −8, −9, −12, and −13. Upon protracted injury, fibrogenesis dominates over fibrolysis, resulting in excess ECM deposition, which is accompanied by a downregulation of MMP secretion and activity, and by an increase of the tissue inhibitors of MMPs (TIMPs), prominently TIMP-1, the major physiological inhibitor of most MMPs which is produced by the majority of liver cell types.

    We have rephrased it to “Myofibroblasts (MFs) are absent in a healthy liver, when the liver is injured, myofibroblasts are activated, which results in highly collagen synthesis and extracellular matrix protein (ECM) accumulation, mainly via down-regulation of matrix metalloproteinases (MMPs, such as MMP-1, -3, -8, -9, -13) and up-regulation of the tissue inhibitors of MMPs (TIMPs, such as TIMP1, TIMP2).”

  1. (Lines 201-202): Unintelligible phrase. Consider rephrasing. 

A: Thank you for pointing this out. As illustrated in “Cell type-specific pharmacological kinase inhibition for cancer chemoprevention,” In a rat model of cirrhosis-driven liver cancer, a small molecule epidermal growth factor receptor inhibitor, erlotinib, was delivered specifically to myofibroblasts by a versatile nanoparticle-based system, targeting platelet-derived growth factor receptor-beta uniquely expressed on their surface in the liver. With systemic administration of erlotinib, tumor burden was reduced to 31%, which was further improved to 21% by myofibroblast-targeted delivery even with reduced erlotinib dose (7.3-fold reduction with equivalent erlotinib dose) and less hepatocyte damage. Erlotinib is a small molecule epidermal growth factor receptor inhibitor, the erlotinib-loaded MFs targeting nanoparticles (PPB-NP-erlotinib) were tested in a cirrhosis-induced liver cancer model. PPB-NP-erlotinib treatment exhibited lower dose erlotinib, while the greater effect in anti-fibrosis.

    We have rephrased it to “To improve the myofibroblasts targeting, erlotinib, a small molecule epidermal growth factor receptor inhibitor, loaded with platelet-derived growth factor receptor b (PDGFRB)-targeting peptide-nanoparticles (PPB-NP-erlotinib). And PPB-NP-erlotinib treatment exhibited a lower dose of erlotinib, while a greater effect in anti-fibrosis.

  1. (Lines 244-245): “Additionally, ECM1 is downregulated in liver fibrosis, which made it a potential biomarker in clinical.”: Incomplete sentence. Consider change.

A: Thank you for pointing this out. We have rephrased it to “The therapeutic efficacy of ectopic ECM1 expression in the chronic CCl4 liver fibrosis mouse model was tested, and the results indicated that ECM1 was successfully expressed in the liver upon AAV8 infection and partially rescued the CCl4 injury-mediated fibrosis. The results strongly support the application of drugs that target TGF-b signaling pathway as a promising strategy to treat liver fibrosis in patients.”

  1. (Line 390): Please explain what is the meaning of the “uncle receptors”.

A: Thank you for pointing this out. We have rephrased it to “nuclear receptor.”

  1. (Line 440): “NTC04171765, recruiting".

A: Thank you for pointing this out. We have made corrections according to your suggestion.

  1. (Line 470): "benzylisoquinoline".

A: Thank you for pointing this out. We have made corrections according to your suggestion.

Reviewer 2 Report

The title of the article fully reflects the content of the article.

The abstract summarizes the necessary information for the reader: the etiology and pathogenesis of liver fibrosis, targets and potential approaches to the treatment of the disease.

The presented keywords are necessary and reflect the research topic presented by the authors.

In the "Introduction" section, the authors briefly, but clearly enough for the reader, presented the main issues of the pathogenesis of liver fibrosis, pointed out the lack of currently approved FDA and EMA methods for the direct treatment of liver fibrosis. The section presents potential targets for drugs for the treatment of liver fibrosis. It is indicated that the most promising approach of therapy may be a combination therapy aimed at two or more strategies. The purpose of this article is clear.

In the following sections of the article, the authors examined the pathogenetic mechanisms of liver fibrosis and the signaling pathways involved in liver fibrosis, as well as various molecular targets for the treatment of the disease. The article discussed the obstacles and opportunities that arise in the development of methods for the treatment of fibrotic liver diseases.

In the section "Conclusion and future perspectives", the authors summarized their analysis of the published data and identified the main issues of pathogenesis and treatment of liver fibrosis that need to be addressed.

It is important that the authors provided Abbreviations separately. It's easy to read.

The table is clear and legible, necessary to understand the content of the article. The figures complement the article, reflect the molecules and mechanisms of liver fibrogenesis, key signaling pathways in liver fibrogenesis, therapeutic possibilities for blocking the development of liver fibrosis.

The article does not cause any concerns. The manuscript did not cause any ethical problems. All references to publications presented by the authors in the article are necessary and correct, made in the right style. Of the 88 references presented in the article, 66 references have been made in the last 5 years (2017-2022). I have no concerns about the similarity of this article with other articles published by the same authors.

Competing interests of authors do not create bias in the presentation of results, analysis and discussion.

Author Response

Dear reviewer, thank you for your appreciation and provided insightful comments about our manuscript. It’s our pleasure to share this review with you. We have made some modifications according to the other two reviewers, and the latest version has been submitted.

Reviewer 3 Report

It is an excellent review that describes the molecular mechanisms of liver cirrhosis and the possibility of treatment based on them. Since the paper is already long enough, it is enough even now, but if possible, if you could mention about half a page about diagnostic markers for liver cirrhosis (markers for progress from chronic hepatitis, etc.) based on molecular basis, It will be helpful to the reader (please be careful not to write too long).

Author Response

Dear reviewer, thank you for your appreciation and provided insightful comments about our manuscript. It’s our pleasure to share this review with you, please see below, in red, for the response to your comments and concerns.

Comments and Suggestions for Authors

It is an excellent review that describes the molecular mechanisms of liver cirrhosis and the possibility of treatment based on them. Since the paper is already long enough, it is enough even now, but if possible, if you could mention about half a page about diagnostic markers for liver cirrhosis (markers for progress from chronic hepatitis, etc.) based on molecular basis, It will be helpful to the reader (please be careful not to write too long).

A: Dear reviewer, thank you for your appreciation and provided insightful comments about our manuscript.

    We have added “Actually, the existing diagnosis methods limit the drug development of liver fibrosis to a certain extent. The diagnostic methods of liver fibrosis can be divided into three categories: liver biopsy, biomarkers, and imaging studies. Although liver biopsy has many limitations, including sampling bias and subjective scoring, it is currently the gold standard in clinical studies. At present, blood biomarkers used in clinical mainly include ALT, AST, and LDL, HDL, TG, which can be detected in hyperlipidemia. Emerging blood biomarkers included ELF, ProC3, and NIS4. ELF is currently used to assist in the diagnosis of patients with NASH with F3 and F4 fibrosis. Imaging tests mainly include ultrasound, Fibroscan, and MRI(MRE, PDFF, cT1). Among these, ultrasound is commonly used in clinical practice. Fibroscan assesses the severity of liver fibrosis by measuring the degree of elasticity of the liver. The advantage of MRI (MRE, PDFF, cT1) over ultrasound and fibroscan is that the detection of the liver as a whole reduces the bias caused by sampling, but the price is expensive relatively. The conduct of clinical trials may require comprehensive diagnosis, and in any case, the development of non-invasive biomarkers is urgent for clinical applications.”
